# Engineering the Steroid Hydroxylating System from *Cochliobolus lunatus* in *Mycolicibacterium smegmatis*

**DOI:** 10.3390/microorganisms9071499

**Published:** 2021-07-13

**Authors:** Carmen Felpeto-Santero, Beatriz Galán, José Luis García

**Affiliations:** Centro de Investigaciones Biológicas Margarita Salas, Agencia del Consejo Superior de Investigaciones Científicas, 28040 Madrid, Spain; carmenfs85@hotmail.com (C.F.-S.); bgalan@cib.csic.es (B.G.)

**Keywords:** steroids, 14α-hydroxylation, *Mycocilibacterium smegmatis*, steroids biotechnology, steroids functionalization, hydroxylated steroidal synthons

## Abstract

14α-hydroxylated steroids are starting materials for the synthesis of contraceptive and anti-inflammatory compounds in the steroid industry. A synthetic bacterial operon containing the cytochrome P450 CYP103168 and the reductase CPR64795 of the fungus *Cochliobolus*
*lunatus* able to hydroxylate steroids has been engineered into a shuttle plasmid named pMVFAN. This plasmid was used to transform two mutants of *Mycolicibacterium smegmatis* named MS6039-5941 and MS6039 that accumulate 4-androstene-3,17-dione (AD), and 1,4-androstadiene-3,17-dione (ADD), respectively. The recombinant mutants MS6039-5941 (pMVFAN) and MS6039 (pMVFAN) were able to efficiently express the hydroxylating CYP system of *C.*
*lunatus* and produced in high yields 14αOH-AD and 14αOH-ADD, respectively, directly from cholesterol and phytosterols in a single fermentation step. These results open a new avenue for producing at industrial scale these and other hydroxylated steroidal synthons by transforming with this synthetic operon other *Mycolicibacterium* strains currently used for the commercial production of steroidal synthons from phytosterols as feedstock.

## 1. Introduction

Steroids represent one of the most widely used compounds as pharmaceuticals such as anti-inflammatory, immunosuppressive, anti-allergic, and anti-cancer drugs [1]. The oxidation state of the steroid nucleus rings and the presence of different attached functional groups determine the biological functions of steroid molecule [2]. Structural modifications of steroids such as hydroxylation, dehydrogenation, and esterification highly affect their biological activity [3,4]. Among them, hydroxylation is one of the most important modifications for steroid functionalization since it introduces deep changes in their physicochemical and pharmaceutical properties (e.g., bioactivity, solubility, and adsorption). Some fungi have been reported to carry out hydroxylations at almost all stereogenic centers of the steroid molecule [5] and they have been widely applied for the industrial production of steroids [6]. Oxyfunctionalization of steroids is mainly catalyzed by cytochromes P450 (CYPs) [7] and in fungi is currently carried out by two-component systems consisting of a NAD(P)H reductase (CPR) and a monooxygenase [5,8].

The hydroxyl group in position 11β of steroids provides important pharmacological properties [9,10] and from a long time this specific hydroxylation has been established as a large-scale commercial process using the fungal CYP of *Cochliobolus lunatus* [11,12,13]. We have identified the genes encoding the cytochrome CYP103168 and the reductase CPR64795 of *C. lunatus* responsible for the 11β-hydroxylase activity in this fungus [14]. We have demonstrated that a recombinant *Corynebacterium glutamicum* strain harboring a plasmid expressing both genes forming a synthetic bacterial operon was able to 11β-hydroxylate several steroids as substrates [14]. In this sense, the design of alternative microbial cell factories created by recombinant DNA technologies has been explored as a challenge in the industry of steroids [6,15,16]. The heterologous co-expression of hydroxylating CYPs and their natural redox partners as putative industrial biocatalyzers has been addressed in different host such as yeasts (e.g., *Pichia pastoris*, *Saccharomyces cerevisiae*, and *Schizosaccharomyces pombe*), and bacteria (e.g., *Escherichia coli*, *Bacillus subtilis*, *Bacillus megaterium, Corynebacterium glutamicum*) [10,14,17,18,19,20,21,22,23,24,25,26,27,28,29,30,31]. All these bioconversion processes are based in whole-cell biocatalysts and require previously purified steroid synthons as substrates. However, it would be interesting to hydroxylate steroids directly from sterols in a single step process. Actinobacteria (e.g., *Mycobacterium*, *Mycolicibacterium*, *Gordonia, Rhodococcus*) are able to metabolize cholesterol and other natural sterols and have been used in the industry to obtain the key steroid synthons (e.g., AD, ADD, 9OHAD) and C22 steroids (e.g., 4-HBC, 1,4-HBC, 9OH-4-HBC) from natural sterols such as phytosterols (Figure 1) [6,32,33,34]. Therefore, these strains could be useful to engineer artificial pathways to achieve in a single step novel hydroxylated steroid synthons. In this sense, we have engineered the 11α-hydroxylating system from the fungus *Rhizopus oryzae* into *Mycolicibacterium smegmatis* showing that the introduction of the fungal system allowed production of 11αOH-AD and 11αOH-ADD from sterols in a single fermentation step [16].

These results prompted us to investigate the possibility of producing 11β-hydroxylated steroids from sterols in *M. smegmatis* using the CYP-CPR system of *C.*
*lunatus.* However, it has been reported that 11β-hydroxylation of steroids carried out by the fungus *C. lunatus* depends of the substrate and thus, 14α- and 7α-hydroxylated steroids can also be obtained [23,35,36,37,38,39]. 14α-Hydroxysteroids are produced primarily using microorganisms and have been used for a long time as precursors for the synthesis of many steroidal drugs [40]. 14α-Hydroxy steroids are starting materials for the synthesis of contraceptive and anti-inflammatory compounds simplifying the chemical pathways for preparing them [38]. Moreover, the configuration of the 14α-hydroxy group can be inverted into the 14β-configuration, which is a structural feature of cardio-active steroids [40,41]. Thus, the heterologous expression of the hydroxylating system of *C. lunatus* in *M. smegmatis* open interesting possibilities and questions about the nature of the resulting hydroxylated synthons that could be obtained once it has been integrated in the catabolism of natural sterols in this strain.

Here, we show that a synthetic operon containing the CYP103168 and its natural partner CPR64795 of *C. lunatus* can be engineered in *M. smegmatis* resulting in the production of 14α-hydroxylated synthons directly from sterols in a single fermentation step.

## 2. Materials and Methods

### 2.1. Chemicals

Commercial phytosterols from pine (provided by Gadea Biopharma, Leon, Spain) containing a mixture of different sterols (w/w percentage): β-sitosterol (83.61%), stigmasterol (8.79%) and campesterol (7.59%), 22-hydroxy-23,24-bisnorchol-4-en-3-one (4-HBC > 98%) and 3-oxo-23,24-bisnorchol-1,4-dien-22-oic acid (1,4-HBC > 97%) were kindly given by Gadea Pharma S.L., 4-androstene-3,17-dione (AD > 99.0%), 1,4-androstadiene-3,17-dione (ADD > 98.0%), and testosterone (TS > 98.0%) were purchased from TCI America (Portland, Oregon, USA). Cholesterol (CHO > 99.0%) and 11β-hydroxy-4-androstene-3,17-dione (11 β OH-AD > 98.0%) were purchase from Merck (Darmstadt, Germany). 14 α-hydroxy-4-androstene-3,17-dione (14αOH-AD > 98.0%) was purchased from Akos GmbH (Stuttgart, Germany).

### 2.2. Strains, Oligonucleotides and Culture Growth

The strains, plasmids, and oligonucleotides used in this study are listed in Table 1. *Escherichia coli* DH10B strain was used as a host for cloning. It was grown in rich LB medium at 37 °C in an orbital shaker at 200 rpm. LB agar plates were used for solid media. Gentamicin (10 μg mL^−1^), ampicillin (100 μg mL^−1^), or kanamycin (50 μg mL^−1^) were used for plasmid selection and maintenance in this strain. *M. smegmatis* mc^2^155 was cultured on Bacto Middlebrook 7H9 (7H9, Difco) supplemented with Middlebrook ADC Enrichement (ADC, Difco) (10% *v*/*v*), glycerol (0.2% *v*/*v*) (Sigma) at 37 °C in an orbital shaker at 200 rpm. Tween 80% (0.05% *v*/*v*) (Sigma) was added to *M. smegmatis* cultures to avoid cell aggregation. Antibiotics were used where indicated at the following concentrations: kanamycin (20 µg mL^−1^). Cell grow was monitored following OD_600nm_.

### 2.3. Construction of the M. smegmatis Heterologous Strains

The CYP103168 and CPR64795 encoding genes were PCR amplified as described from cDNA of *C. lunatus* [14]. To achieve optimal translation in the host bacterium, the PCR primers contained a Shine Dalgarno sequence (AAAGGGAG) upstream of each gene at 6 bp from the respective start codons (Appendix A). The primers used to perform these amplifications are listed in Table 1. Restriction enzyme sites were also added to the primers to clone the genes into the plasmid vector. The isolated DNA fragment containing the CPR64795 encoding gene were further digested with *Eco*RI and *Xba*I, whereas the fragment containing the CYP103168 encoding gene was digested with *Xba*I and *Xho*I. These fragments were isolated and ligated to the pMV260 shuttle *E. coli/Mycobacterium* plasmid [43] previously digested with *Eco*RI and *Sal*I yielding pMVFAN plasmid carrying the CPR64795 and CYP103168 encoding genes forming a synthetic operon (named FAN operon) expressed under the control of the *P_hsd_* promoter. The pMVFAN plasmid was transformed into *E. coli* DH10B competent cells and sequenced to confirm the accuracy of the construction. Then, plasmids pMV261 (empty vector, control plasmid) and pMVFAN were individually transformed into the *M. smegmatis* mutant strains MS6039 and MS6039-5941 [44] by electroporation.

### 2.4. Steroid Biotransformation Assay

Sterol biotransformation assays were carried out using growing cells of *M. smegmatis* MS6039 (pMVFAN) and MS6039-5941 (pMVFAN) cultured in 100 mL flasks containing 20 mL of biotransformation medium (7H9 Broth supplemented with 18 mM glycerol as starter, 1 mM cholesterol or 1 mM phytosterols (previously dissolved in 3.6% (*v*/*v*) Tyloxapol), 0.5 mM δ-aminolevulinic acid (ALA) and kanamycin (20 µg mL^−1^)). The culture was inoculated with cells that have been pre-cultured for 48 h in Bacto Middlebrook 7H9 (7H9 supplemented with Middlebrook ADC Enrichement (10% *v*/*v*), glycerol (0.2% *v*/*v*), kanamycin (20 µg mL^−1^) and Tween 80 (0.05% *v*/*v*). The pre-culture was centrifuged and washed with one volume of NaCl-Tween 80 solution prior to its inoculation. The pellet was resuspended in 0.5 mL of the washing solution to measure the OD_600_. The biotransformation flasks were inoculated to an initial OD_600_ of 0.05 and cultured on an orbital shaker (250 rpm) at 37 °C during 96 h. Culture samples (1 mL) were taken at 0, 24, 48, 72, and 96 h to monitor cell growth by OD_600_ and sterol products by HPLC-DAD-MS.

### 2.5. Steroid Extraction and Analyses

Aliquots of 10 µL of 5 mM testosterone in 10% (*v*/*v*) Tyloxapol were added to 0.2 mL of biotransformation medium prior to extraction with chloroform, as an internal standard (ISTD). The samples were extracted using two volumes of chloroform. The aqueous fraction was discarded, the chloroform fraction was dried at 60 °C using a Thermoblock and the pellet dissolved in 100 μL of acetonitrile. Each sample was further analyzed by to thin layer chromatography (TLC) and HPLC-UV-DAD-MS.

Then, 10 μL of standards and samples extracted as described above were spotted in silica gel plates (TLC Silica gel 60 F_254_, Merck Millipore, Darmstadt, Germany). Chloroform: ethanol (30:150 *v*/*v*) was used as a developing system. Steroid products were visualized by UV or revealed by spraying 20% (*v*/*v*) sulfuric acid and heating at 100 °C. To purify the AD hydroxylated compound a preparative TLC was performed.

HPLC-UV-DAD-MS analyses were carried out using a DAD detector and a LXQ Ion Trap Mass Spectrometer, equipped with an atmospheric pressure chemical ionization source, electrospray ionization source and interfaced to a Surveyor Plus LC system (all from Thermo Electron, San Jose, CA, USA). Data were acquired with a Surveyor Autosampler and MS Pump and analyzed with the Xcalibur Software (from Thermo Fisher Scientific, San Jose, CA, USA). High-purity nitrogen was used as nebulizer, sheath, and auxiliary gas. MS analysis was performed both in full scan and in selected ion monitoring (SIM) mode by scanning all the daughter ions of the products in positive ionization mode. The quantification was performed from parent mass of compounds, and the specificity was obtained by following the specific fragmentations of all compounds.

The experiments were carried out with the following interface parameters: Ionization source APCI, injected samples of 25 µL, capillary temperature 275 °C, vaporizing temperature 425 °C, capillary voltage 39 V, corona discharge 6 kV, source power 6 µA and dissociation by collision energy 15 eV. Chromatographic separation was performed on a *Tracer Excel* 120 ODSB C18 (4.6 × 150 mm, particle size 5 μm) (Teknokroma, Barcelona, Spain). The chromatography was performed using water containing 0.1% (*v*/*v*) of formic acid, acetonitrile containing 0.1% (*v*/*v*) of formic acid and isopropanol containing 0.1% (*v*/*v*) of formic acid as mobile phases A, B and C, respectively (flow 1 mL min^−1^). To monitor cholesterol (CHO) biotransformations, the analyzed compounds were Testosterone (289 *m*/*z*) used as ISTD (internal standard), CHO (269 *m*/*z*), AD (287 *m*/*z*), 14αOH-AD (303 *m*/*z*), ADD (285 *m*/*z*), 14αOH-ADD (301 *m*/*z*), 4-HBC (331 *m*/*z*), and 1,4-HBC (329 *m*/*z*). The HPLC gradient used was as follows: **Time (min)****%A****%B****%C**05050055050015207192048794008515410851542505005250500

Quantification of the compounds was calculated by the reaction yield and peak area regarding the ISTD.

To monitor the phytosterol (PHYTHO) biotransformation that is a mixture of sitosterol (SITO), stigmasterol (STIG), and campesterol (CAMP), the analyzed compounds were: Testosterone (289 *m*/*z*) used as ISTD, CAMP (383 *m*/*z*), STIG (395 *m*/*z*); SITO (397 *m*/*z*), AD (287 *m*/*z*), 14αOH-AD (303 *m*/*z*), 14αOH-AD (303 *m*/*z*), ADD (285 *m*/*z*), 14αOH-ADD (301 *m*/*z*), 4-HBC (331 *m*/*z*) and 1,4-HBC (329 *m*/*z*). The HPLC gradient used was as follows:
**Time (min)****%A****%B****%C**05050055050015207192009194007030410851542505005250500

The concentration of sterols in 1 mM PHYTO is as follows: 0.84 mM SITO + 0.08 mM STIG + 0.08 mM CAMP. Therefore, PHYTO represents the sum of the amounts of SITO, STIG and CAMP. The quantification of the compounds was calculated by the reaction yield and peak area regarding ISTD.

### 2.6. NMR Spectra

The NMR spectra were acquired at 298 K in Bruker AVANCE 500 MHz or 600 MHz spectrometers. All the samples were dissolved in deuterated chloroform. The RMN spectrum of the sample purified by TLC was compared with those of standard commercial samples of 11βOH-AD and 14αOH-AD. One dimensional ^1^H and 2D heteronuclear ^1^H-^13^C HSQC and DOSY (Diffusion Ordered Spectroscopy) spectra were acquired using the corresponding pulse sequences included in TOPSIN acquisition and spectral analysis software from Bruker.

## 3. Results and Discussion

Current methods of production of hydroxylated steroids mainly rely on biotransformations using wild-type fungal whole cells that harbor these enzymatic activities. The production of the hydroxylated steroids is carried out in at least two fermentation steps exhibiting in most cases some drawbacks such as low selectivity and reduced conversion yield. Therefore, the design of alternative fermentation processes by using recombinant DNA technologies has been proposed as a challenge in recent years [1]. Thus, to advance in this direction, this is to develop alternative processes for producing hydroxylated steroids directly from natural sterols in a single fermentation step, we decided to test the effects of the expression of CYP103168 of *C. lunatus* in different *M. smegmatis* mutants that produced AD or ADD from natural sterols.

### 3.1. Production of 14αOH-AD in M. Smegmatis from Sterols in Single Fermentation Step

The *M. smegmatis* MS6039-5941 mutant strain, engineered to produce AD from CHO or PHYTO [33,44], was used as bacterial chassis to study the conversion of natural sterols into hydroxylated steroids by the action of the hydroxylating system of *C. lunatus* (Figure 2).

Firstly, the recombinant MS6039-5941 (pMVFAN) and MS6039-5941 (pMV261) (control plasmid) strains were cultured in the presence of CHO. Curiously, an increased autolysis was observed at stationary phase of MS6039-5941 (pMVFAN) when compared to the control strain, probably due to some toxic effect caused by CYP-CPR activity (Figure 3A). The maximum bioconversion of 99.8 ± 0.3% was achieved after 48 h of growth. The product yield obtained was 61.4 ± 1.2% OH-AD/CHO (Figure 3B). Besides the OH-AD, other products as AD (AD/CHO = 28.2 ± 1.0%), 4-HBC (4-HBC/CHO = 4.5 ± 0.6%) and OH-ADD (OH-ADD/CHO = 5.9 ± 0.4%), were produced during the biotransformation (Figure 3B and Figure 4).

Trace amounts of four additional compounds were detected (Figure 4). The unidentified compound eluting at 8.1 min can be assigned based on its *m*/*z* of 347, polarity, and fragmentation pattern to OH-4-HBC. Two compounds co-eluting at 5 min with *m*/*z* of 347 and 303 were also detected that could be 4 HBC and AD hydroxylated derivatives in different positions, respectively. Another compound eluted at 6.2 min with *m*/*z* of 345 could be the 1,4-HBC hydroxylated derivate.

It is worth to mention that the control strain MS6039-5941 (pMV261) did not produce OH-AD (Appendix A) and as expected, AD is the main product, having a yield of 76.7 ± 3.7% with a conversion of 99.1 ± 0.2%. Curiously, a significant amount of 4-HBC was detected in the control strain (Y_4-HBC/CHO_ = 18.3 ± 2.9%) when compared to the strain carrying the FAN operon. A small amount of ADD was also detected in the control strain (Y_ADD/CHO_ = 4.9 ± 0.9%).

At this point it is worth to mention that CYP103168 is a 60-kDa microsomal membrane-bound protein that was purified and described as a bifunctional enzyme because it catalyzes both steroid 11β- and 14α-hydroxylation reactions in a molar ratio of about 2:1 when cortexolone (Reichstein’s compound S, RSS) is used as substrate, but catalyzes the 14α-hydroxylation of AD [36]. When the *C. lunatus* VKPM F-981 strain was used for AD biotransformation, the major product was identified as 14αOH-AD while the 11βOH-AD by-product was only present in trace amounts [38,39]. The steroid 14α-hydroxylation activity of this CYP towards RSS and AD has been also tested heterologously in *S. cerevisiae* where they showed a specific 14α-hydroxylase activity towards the AD substrate (regiospecificity > 99%); but a poor C14-hydroxylation regiospecificity (around 40%) for the RSS substrate [23].

Considering that *C. lunatus* can lead to the formation of 14α-OH steroids depending on the substrate used for the biotransformations [23,37] we elucidated by NMR the structure of the OH-AD produced by the MS6039-5941(pMVFAN) strain. Based on its positive ion mass spectrometry (MS) [M + Na]^+^ result at *m*/*z* 305.1955 (calculated value, 305.2111) [45], the molecular formula of this product was determined to be C19H26O3 that was compatible with both 11βOH-AD and 14αOH-AD. It is worth to mention that both compounds elute at the same retention time in HPLC and have the same MS fragmentation spectra (Appendix A data not shown). Therefore, the unique method to identify the product was NMR. To this aim, we compared the NMR spectra of the OH-AD product purified from a TLC plate with the standard samples of 11βOH-AD and 14αOH-AD. The absence of the signal at δ 4.6 in the purified sample excluded the possibility that the synthesized compound could be 11βOH-AD. On the other hand, the correlation of the NMR signals of the purified sample with those of the 14αOH-AD standard strongly suggested that the new synthesized compound is 14αOH-AD (Appendix A).

Once determined that the main product derived from cholesterol was 14αOH-AD, we checked the ability of the MS6039-5941 (pMVFAN) and MS6039-5941 (pMV261) strains to convert PHYTO into 14αOH-AD. The increased autolytic effect at the stationary phase of the recombinant carrying the CYP-CPR system compared to the control strain is also observed in this biotransformation medium (Figure 3C). The HPLC-DAD-MS monitoring revealed that PHYTO was transformed into 14αOH-AD (Figure 3D and Appendix A) reaching a conversion of 69.3 ± 2.9% at 96 h. At this time, the product yield was 65.1 ± 2.2% for 14αOH-AD/PHYTO. In addition, other products, such as AD (Y_AD/PHYTO_ = 23.0 ± 2.5%), 4-HBC (Y_4-HBC/PHYTO_ = 3.9 ± 0.1%) and 14αOH-ADD (Y_14αOH-ADD/PHYTO_ = 6.5 ± 2.0%) were detected in the culture supernatant (Figure 3D and Appendix A).

As expected, the MS6039-5941 (pMV261) control strain did not render any 14αOH-AD and produced AD achieving a conversion of 52.7 ± 12.9% yielding 77.6 ± 2.0% AD/PHYTO. A significant amount of 4-HBC (Y_4-HBC/PHYTO_ = 17.0 ± 1.2%) and a small amount of ADD (Y_ADD/PHYTO_ = 3.3 ± 0.4%) were produced as by-products.

### 3.2. Production of 14αOH-ADD in M. smegmatis from Sterols in Single Fermentation Step

To test the possibility of producing large amounts of 14αOH-ADD from natural sterols, the plasmid pMVFAN carrying the 11β-hydroxylating system of *C. lunatus* was transformed into *M. smegmatis* MS6039, a mutant strain that accumulates ADD from sterols (Figure 2) [33,44]. As a control, we used the same strain transformed with the empty plasmid pMV261. The production of 14αOH-ADD was analyzed by HPLC-DAD-MS along the growth curve of MS6039 (pMVFAN) and MS6039 (pMV261) using sterols as feedstock.

Firstly, MS6039 (pMVFAN) and MS6039 (pMV261) strains were grown in the in the presence of CHO. The growth curves of both strains were very similar although an increased autolytic effect can be observed at stationary phase in the MS6039 (pMVFAN) suggesting some toxicity (Figure 3E). The maximum bioconversion of 99.9 ± 0.1% was observed after 96 h of growth with a 14αOH-ADD production yield of 31.3 ± 0.1% (Figure 3F). Besides 14αOH-ADD other products were detected in the biotransformation assay such as ADD, AD, and 14αOH-AD (Figure 3F and Figure 5) The production yields of the main products were 61.9 ± 0.1% for ADD/CHO and 6.8 ± 0.1% for 14αOH-AD/CHO. 1,4-HBC and AD were present at very low concentrations. One additional compound was detected co-eluting at 4 min with 14αOH-AD (Figure 5 and Appendix A). Based on its *m*/*z* 345, it could be assigned to a hydroxylated derivative of 1,4-HBC.

As expected, the MS6039 (pMV261) control strain did not produce 14αOH-ADD from CHO (Appendix A) and ADD was detected as the main biotransformation product with a conversion rate of 96.2 ± 5.9% and a production yield of 99.3 ± 0.3% for ADD/CHO. Trace amounts of 1,4-HBC were detected as well (Appendix A).

Secondly, considering that PHYTO is used as substrate in the steroid industry as the preferred low-cost raw material to produce steroid synthons, we tested PHYTO as feedstock to produce 14αOH-ADD in the MS6039 (pMVFAN) recombinant strain. To this aim, MS6039 (pMVFAN) and MS6039 (pMV261) strains were grown in the presence of PHYTO (Figure 3G) and the biotransformation was monitored by HPLC DAD-MS. Figure 3H shows that the MS6039 (pMVFAN) strain successfully achieved the transformation of PHYTO into 14αOH-ADD. The conversion rate was 70.9 ± 1.4% and the 14αOH-ADD production yield 14αOH-ADD/PHYTO was 31.1 ± 5.5%. Some by-products as ADD, 14αOH-AD, 1,4-HBC and a compound of *m*/*z* 345 compatible with OH-1,4-HBC were detected when PHYTO was used as feedstock (Appendix A). The yields for these products were 62.0 ± 8.4% for ADD/PHYTO and 4.3 ± 1.3% for 14αOH-AD/PHYTO. 1,4-HBC and the possible OH-1,4-HBC were present at very low concentrations.

As expected, MS6039 (pMV261) control strain only produced ADD from PHYTO with a conversion rate of 67.5 ± 0.3% and a transformation yield of 95.7 ± 0.9% for ADD/PHYTO. A small amount of 1,4-HBC was detected.

Our results demonstrate that fortunately, the synthetic operon FAN was fully functional in *M. smegmatis* since only few fungal CYPs have been successfully produced in its active form in bacteria so far [14,16,24,25,29]. The detection of 14α-hydroxylated AD and ADD in the culture medium of the recombinant bacteria at high yields demonstrated that the fungal hydroxylating system has been integrated in the sterol metabolism of *M. smegmatis* creating a new expanded pathway. To our knowledge, the production of 14α-hydroxysteroids in mycobacterial cells has not previously been achieved.

Our findings also suggest that CYP103168 does not efficiently hydroxylate in *M. smegmatis* other C21 intermediates derived from sterol catabolism, such a 4-HBC and 1,4-HBC that still retain part of the sterol side chain. Although we have observed some products that might correspond to the hydroxylated derivatives of 4-HBC and 1,4-HBC, they appear in minimal amounts. Taking into account that we have shown that CYP103168 is able produce 11βOH derivatives of steroids with a side chain in *C. glutamicum* [14] it could be expected that 4-HBC and 1,4-HBC could be mainly hydroxylated in C11. If this was case the final hydroxylated products should be 11βOH-AD and 11βOH-ADD or a mixture of 11βOH and 14αOH products. The fact that we detected only the 14αOH products suggests that CYP is using as substrates mainly the final products AD and ADD and not the sterol intermediates. This can be explained because the transformation of sterols into AD and ADD is very efficient and the intermediates are accumulated only at a low concentration far from the Km of CYP. Thus, only when AD and ADD are accumulated at high concentrations they become substrates of CYP. However, we cannot exclude the possibility that 4-HBC and 1,4-HBC could be very poor substrates of CYP, since these substrates have not been tested so far. To test these hypotheses, it could be possible to determine the behavior of CYP in *M. smegmatis* mutants that accumulate 4-HBC and 1,4-HBC, but these mutants have not been created yet.

Although the whole process must be further optimized at industrial scale to improve the yield of the 14α-hydroxylated compounds by increasing the substrate consume and reducing the amount and number of by-products, our results open a new avenue for searching effective biocatalysts for producing hydroxylated steroids from sterols in a single fermentation step. In addition, they reinforce the assumption that engineered *M. smegmatis* strains represent a new generation of biocatalysts with a great potential to be applied for industrial processes. Based on the increased knowledge on the steroid metabolism in *M. smegmatis,* we uphold this bacterium as an exceptional bacterial chassis to implement *à la carte* metabolic engineering strategies based on synthetic biology for the industrial production of other valuable pharmaceutical steroids directly from sterols.

## Figures and Tables

**Figure 1 microorganisms-09-01499-f001:**
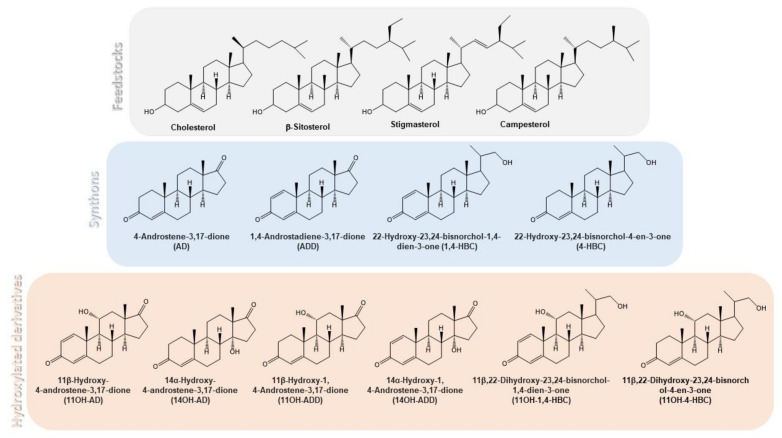
Chemical structure of the most relevant steroidal compounds described in this work.

**Figure 2 microorganisms-09-01499-f002:**
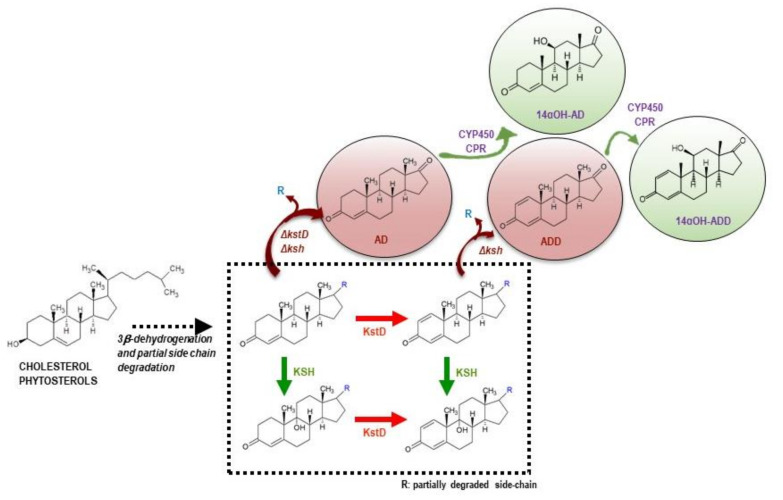
Scheme of the rational construction of mycobacterial strains producing hydroxylated steroidal intermediates from natural sterols by incorporating the hydroxylating system from *C. lunatus*. *kstD*, 3-ketosteroid-Δ^1^-dehydrogenase; *ksh*, 3-ketosteroid-9α-hydroxylase.

**Figure 3 microorganisms-09-01499-f003:**
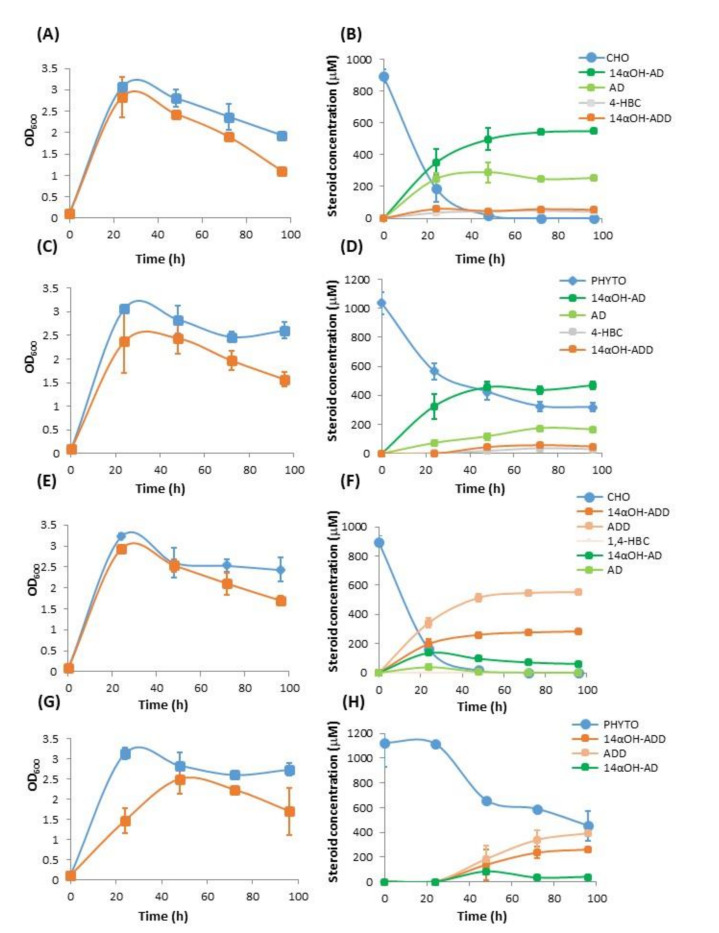
Biotransformation assays in the presence of sterols (CHO or PHYTO) performed with the MS6039-5941 (pMVFAN) and MS6039 (pMVFAN) recombinant strains. Results represent means of three biological replicates. Error bars represent the standard deviation. (**A**) Growth curves of MS6039-5941 (pMVFAN) (red) and MS6039-5941 (pMV261) (blue) strains in the biotransformation medium in the presence of CHO. (**B**) Consumption of CHO and steroidal biotransformation products delivered by MS6039-5941 (pMVFAN). (**C**) Growth curves of MS6039-5941 (pMVFAN) (red) and MS6039-5941 (pMV261) (blue) strains in the biotransformation medium in the presence of PHYTO. (**D**) Consumption of PHYTO and steroidal biotransformation products delivered by MS6039-5941 (pMVFAN). (**E**) Growth curves of MS6039 (pMVFAN) (red) and MS6039 (pMV261) (blue) strains in the biotransformation medium containing CHO. (**F**) Consumption of CHO and steroidal biotransformation products delivered by MS6039 (pMVFAN). (**G**) Growth curves of MS6039 (pMVFAN) (red) and MS6039 (pMV261) (blue) in the biotransformation medium containing PHYTO. (**H**) Consumption of PHYTO and steroidal biotransformation products delivered by MS6039 (pMVFAN).

**Figure 4 microorganisms-09-01499-f004:**
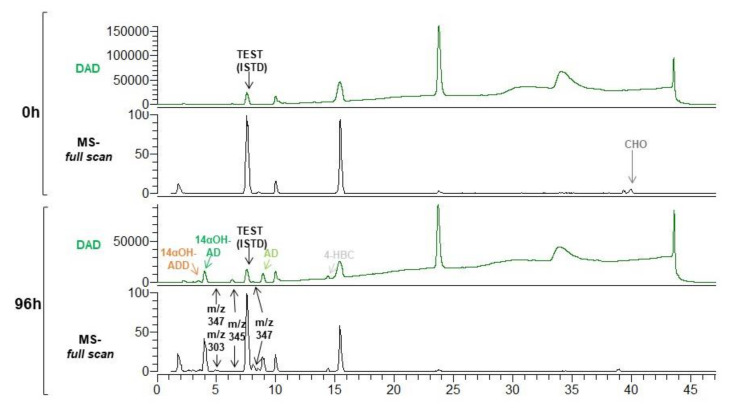
CHO biotransformation by MS6039-5941 (pMVFAN) strain. HPLC-DAD chromatogram (50–600 nm) (green) and and full scan mass spectra (*m*/*z* 150-400) at 0 h and 96 h of growth are shown.

**Figure 5 microorganisms-09-01499-f005:**
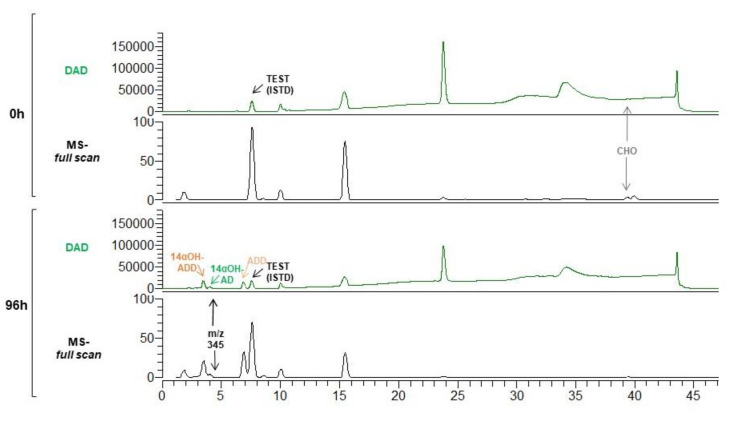
CHO biotransformation by MS6039 (pMVFAN) strain. HPLC-DAD chromatogram (50–600 nm) (green) and full scan mass spectra (*m*/*z* 150–400) at 0 h and 96 h of growth are shown.

**Table 1 microorganisms-09-01499-t001:** Strains, plasmids and oligonucleotides used in this work.

Strain	Description	Reference
*Mycolicibacterium smegmatis* mc^2^ 155	ept^−1^, mutant mc^2^6	[42]
*Mycolicibacterium smegmatis* mc^2^ 155 MS6039	*M. smegmatis* mc^2^ 155 Δ*MSMEG_6039*	[34]
*Mycolicibacterium smegmatis* mc^2^ 155 MS6039-5941	*M. smegmatis* mc^2^ 155 Δ*MSMEG_6039*-Δ*MSMEG_5941*	[34]
*Escherichia coli* DH10B	F^−^, *mcrA, Δ(mrr-hsdRMS-mcrBC), f80ΔlacZDM15 ΔlacX74, deoR, recA1, endA1, araD139, Δ(ara-leu)7697, galU, galK, rpsL, nupG, λ^−^*	Invitrogen
**Plasmid**	**Description**	**Reference**
pMV261	*Km^R^, Mycobacterium* expression vector, *P_Hsp60_*	[43]
pMVFAN	*Km^R^*, synthetic operon FAN into pMV261	This work
**Oligonucleotide**	**Sequence**	**Application**
pMV4260 F	TTGCCGTCACCCGGTGACC	pMV261 sequencing
pMV4486 R	ATCACCGCGGCCATGATGG	pMV261 sequencing
64795 EcoRIF	CCGGAATTCTGACCTGAGAGAAAGGGAGTGATAAATGGCACAACTCGACACGC	CPR64795 amplification
64795 XbaIR	GCTCTAGATTATCATGACCAGACGTCTTCCTG	CPR64795 amplification
103168 BglXbaMunF	GAAGATCTTCTAGACAATTGTGACCTGAGAGAAAGGGAGTGATAAATGGATCCCCAGACTGTCG	CYP103168 amplification
103168 XhoR	CCGCTCGAGTTACTACACTACCACTCTCTTGAAAGC	CYP103168 amplification
64795 F2	AATCAGCATTGCTGGCTCC	FAN operon sequencing
64795 F3	CTCCAACTTCAAGCTTCCTTCG	FAN operon sequencing
64795 F4	AATACGTCGCTTTCGGTCTCG	FAN operon sequencing

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
