# Peer review of "Engineering the Steroid Hydroxylating System from Cochliobolus lunatus in Mycolicibacterium smegmatis"

_microorganisms, 2021, doi:10.3390/microorganisms9071499_

Round 1

Reviewer 1 Report

Major concerns:

The production of steroidal synthons from cheap precursors such as cholesterol and phytosterols has been carried out in pharmaceutical industry for decades. However, previous bioprocess mainly relied on the application of chemical inhibitors to block the core-ring degradation in the steroid catabolic pathway. This required the continuous involvement of chemicals in bacterial cultures, thus it is not an economical and eco-friendly process. The manuscript written by Felpeto-Santero et al. described the cloning and heterologous expression of the fungal cytochrome P450 CYP103168 (with the 11-hydroxylase activity) and reductase genes into the actinobacterial mutants. The model organism Mycolicibacterium smegmatis is well known for its sterol degradation efficiency and its genome is available. In this study, used as whole-cell biocatalysts, the recombinant mutants MS6039-5941 (pMVFAN) and MS6039 (pMVFAN) were able to efficiently express the hydroxylating CYP system of C. lunatus and produced in high yields 14αOH-AD and 14αOH-ADD, respectively, directly from sterols in a single fermentation step. This study significantly improved our understanding of biotechnological production of steroid drugs. In general, this manuscript is well written and the experiments are well designed. I have only some minor comments:

Minor comments:

  1. Introduction: the authors should clearly describe why genes from fungus Rhizopus oryzae were focused? Why not the homologous genes from other fungi or bacteria?
  2. The bacterial steroid catabolic pathway involved in this manuscript (namely the 9,10-seco pathway) are unfamiliar to most readers. It would be very nice if the author show the pathway, along with the enzymes described in this manuscript, in a figure (Figure 2). By doing so, the readers can easily understand the steroid metabolism by the wild-type and mutants of the actinobacterial model organism as well as how the hydroxysteroids can be produced by the Mycobacterium mutants containing the fungal genes.
  3. Figure 1: The chemical structures of 11α-hydroxysteroids are shown in Fig. 1; however, the given names below are “11-Hydroxy…”, which should be clearly mentioned as “11α-Hydroxy…”. Alternatively, the C-C bond of the 11-hydroxyl group can be shown using a solid line.
  4. Methods (2.4. Steroid biotransformation assay): Unlike sex steroids such as testosterone or E1, the water solubility of sterols is quite low (< 1 mg/L). It may not a good idea to monitor cell growth by OD600 as the insoluble sterol particles may apparently interfere OD 600 values. In this case, the monitoring of bacterial growth, especially in initial stages, can be carried out more exactly by determining the increase in bacterial proteins in cultures.
  5. Methods and Results (Construction of the M. smegmatis heterologous strains): The text regarding the cloning and construction of bacterial mutants are difficult to read. In addition, the data regarding the confirmation the gene transformation and expression are lacking. It would be very nice if the authors can show the plasmid design and the genotype confirmation in a figure.
  6. There are currently only two figures in the manuscript. Some chromatograms and MS spectra relevant to this manuscript (e.g., Figure S1AB) should be moved from SI into the main text.
  7. The experimental data are not much. Thus, I suggest to combine the Results and Discussion together, following by a Conclusion.
  8. Results: The authors described the block of the steroid catabolic pathway and the apparent accumulation of hydroxysteroid products in bacterial cultures. Based on my experience, bacteria tend to excrete end products or bypass products into environments as these compounds may interfere bacterial physiology or metabolism. Alternatively, high concentrations of 1,4-diene steroids are toxic to bacterial cells as they may interfere membrane structure. I was wondering if the authors have observed the extracellular distribution of these hydroxysteroids? The extracellular steroids can be separated from bacterial cells easily, which may facilitate the large-scale purification of these steroid products.

Reviewer 2 Report

A valuable work, which can be improved and I hope the comments help to do so:

Abstract provides motivation at the last section and gives results before; rephrase to a proper order: Background, motivation, methods, results, and conclusion.

There are hyperlinks in the abstract. Those should be removed and if considered to be relevant place them in methods section where you mention the strain as a source of gene.

Figure 1: improve quality! Further you may not only introduce the structure, name and abbreviation but also the potential of industrial uses next to it.

In the methods you have still sigma or sigma-aldrich listed; you may want to redirect to merck now as this might be easier for those repeating this with similar research questions.

In results section you have some data not shown but those might be important; may add the information to supplemental at least.

Some strange symbols are given along with the compound designations and those should be corrected towards proper designations.

Quality of Fig 2 need to be improved. Scale should be in all cases similar; e.g. 1200 µM substrate conc ... in order to allow a direct comparision

References need to be corrected for style (capital letters, Journal abbreviations etc.!
